# Effects of Different Iron Supplements on Reproductive Performance and Antioxidant Capacity of Pregnant Sows as Well as Iron Content and Antioxidant Gene Expression in Newborn Piglets

**DOI:** 10.3390/ani13030517

**Published:** 2023-02-01

**Authors:** Xiaokun Xing, Chunyong Zhang, Peng Ji, Jia Yang, Qihua Li, Hongbin Pan, Qingcong An

**Affiliations:** Yunnan Provincial Key Laboratory of Animal Nutrition and Feed Science, Faculty of Animal Science and Technology, Yunnan Agricultural University, Kunming 650201, China

**Keywords:** pregnant sow, piglet, iron supplement, lactoferrin, heme-iron, iron-glycine complex, productivity, oxidation resistance

## Abstract

**Simple Summary:**

Anemia in piglets is a common problem in piglet production. In this experiment, three different iron additives were used in pregnant sow feed for comparison. The results revealed that adding iron supplements can improve the antioxidant capacity of sows, and lactoferrin and heme iron can improve the iron nutrition and antioxidant capacity of piglets. This study should help improve the reproductive performance of sows, prevent iron deficiency anemia in piglets, and improve disease resistance.

**Abstract:**

To improve the reproductive performance of sows and the iron nutrition of newborn piglets, we studied the effects of dietary iron on reproductive performance in pregnant sows as well as antioxidant capacity and the visceral iron content of sows and newborn piglets. Forty pregnant sows were divided into four groups, the iron deficiency group (Id group) was fed a basic diet while sows in the treatment groups were fed diets supplemented with 200 mg/kg lactoferrin (LF group), 0.8% heme-iron (Heme-Fe group), or 500 mg/kg iron-glycine complex (Fe-Gly group). The results indicated that (1) different sources of iron had no significant effect on litter size, live litter size, and litter weight of sows; (2) the three additives improved iron nutrition in newborn piglets, with LF and Heme-Fe having better improvement effects; and (3) the addition of different iron sources improved the level of serum antioxidant biochemical indexes of sows and newborn piglets, and it can have an effect on gene level, among which lactoferrin has the best effect. Thus, adding LF, Heme-iron, or Fe-Gly to the diet of sows during the second and third trimester of gestation can improve the antioxidant capacity of the sows. The supplementation of LF in pregnant sow diets can also improve the antioxidant capacity and the iron nutrition of newborn piglets, with better additive effects than in Heme-Fe and Fe-Gly.

## 1. Introduction

Iron is a component of hemoglobin and myoglobin and is the prosthetic (heme) group for a variety of enzymes such as cytochrome oxidase, peroxidase, and catalase (CAT) in animals. Iron participates in the transport of oxygen in the body and biological oxidation in cells, and plays an essential role in maintaining hematopoiesis, antioxidation, and immune functions in animals [1]. During pregnancy, animals require additional iron supplementation to meet their own and fetal needs [2]. Common dietary iron additives include amino acid chelated iron, organic acid iron, and inorganic iron [3]. However, due to placental and mammary barriers in sows, ferrous sulfate supplementation does not increase iron storage in piglet livers, and thus newborn piglets also need supplementation [4].

Compared with inorganic trace elements, organic iron can better support the needs of the animal body [5]. Organic acid chelated iron, such as fumarate ions in ferrous fumarate, participates in the tricarboxylic acid cycle and forms adenosine triphosphate (ATP) for body metabolism [6]. Iron amino acid chelates, such as ligand glycine in ferrous glycine, which has an analogous structure to heme, play a necessary role in synthesizing heme as carriers of iron to target cells [7]. The addition of ferrous glycinate chelate to piglet diets can improve iron homeostasis, oxidation, and mitochondrial function in newborn piglets with anemia [8].

Lactoferrin is an important iron transporter that regulates iron balance, antioxidation, and bone marrow cell generation in vivo. Lactoferrin (LF) also exhibits broad-spectrum antibacterial, antiviral, and antitumor functions, as well as immune system regulation [9]. Its molecular structure contains two highly conserved iron-binding sites with a high affinity for Fe^3+^. The iron saturation of LF in the natural state ranges from 10% to 20% [10,11]. LF can also block the production of oxygen free radicals, inhibit lipid peroxidation, and reduce oxidative tissue damage by binding free iron [12]. Several important enzymes also depend on LF and iron ions to exert redox activity. After entering the gastrointestinal tract, proteins are rapidly hydrolyzed into small peptides, which are then absorbed by the intestinal tract. During this time, their original sequence, structure, and physiological function are impacted. However, LF can be absorbed into blood circulation through the immature intestinal tract [13] and can also enter humoral circulation in infants [14]. As a feed additive, LF can have an excellent effect on preventing gastrointestinal tract disorders, improving immune function, and promoting growth in piglets [15].

The absorption of iron in the intestine is divided into heme and non-heme mechanisms. In duodenum, under the action of membrane-bound iron reductase, non-heme iron (Fe^3+^) in the diet is converted into iron (Fe^2+^) and is then absorbed through divalent metal transporters (DMT1), while heme iron is absorbed via heme carrier protein 1 (HCP1). First, hemoglobin in the intestinal cavity is decomposed into heme and globulin degradation products under the action of intestinal enzymes. Heme enters the intestinal mucosal epithelial cells in the form of complete metalloporphyrins. Under heme oxidase activity in the cytoplasm, the porphyrin ring opens, releasing free bivalent iron to enter the non-heme iron pool. Free bivalent iron is involved in the metabolism of iron in cells. The bioavailability of Heme-Fe in plant-based piglet feed is lower than that of animal protein [16]. Thus, dietary iron supplementation can improve the iron state and intestinal development in piglets. However, Heme-Fe exhibits better bioavailability and higher iron deposition in muscle and liver than the traditional additive FeSO_4_ [17].

Few studies have compared the effects of different iron supplements on pregnant sows and the resulting piglets. In the current study, we added LF, Heme-Fe, or Fe-Gly to the diet of pregnant sows and explored the effects of the different iron agents on reproductive performance in sows and on iron content and antioxidant performance in newborn piglets. The results of this study should help improve the reproductive performance of sows, prevent iron deficiency anemia in piglets, and improve disease resistance.

## 2. Materials and Methods

### 2.1. Animal Ethics

All animal care and treatment procedures were carried out in strict accordance with the ethical requirements of the Institutional Animal Care and Use Committee of Yunnan Agricultural University, China. License number: YNAU20180220.

### 2.2. Sow and Rearing Conditions

The experiment adopted a randomized design. In total, 40 healthy pregnant sows (Landrace × Yorkshire) of similar parity (3–4 parities, similar in weight) were divided into four groups equally, with each sow raised separately. Disinfection, immunization, and feeding management procedures were carried out in conformity with the requirements of farm production management. From the 33rd day of pregnancy until delivery, the sows were fed the experimental feed twice daily (7:30 am and 5:30 pm). Pregnant sows were fed 3.5 kg of feed every day.

### 2.3. Diet Preparation

According to the National Research Council (NRC) Nutrient Requirements of Swine (2012) and China Pig Raising Standard (NY-T 65-2004), we prepared a basal diet for the pregnant sows (Table 1) without iron trace element additives. The iron deficiency group (Id group) was fed the basic diet, whereas the treatment groups were fed the basal diet supplemented with 200 mg/kg lactoferrin (LF group), 0.8% heme iron (Heme-Fe group), or 500 mg/kg iron-glycine complex (Fe-Gly group), and the iron available to sows in the treatment groups was similar. Lactoferrin was obtained from the DMV International Co., Ltd. (Veghel, The Netherlands), Protein 98.7%, Lactoferrin 93.3%, Iron Saturation 10.1%. The Futiebao iron-glycine complex (glycine ≥ 21% and Fe^2+^ ≥ 17%) was purchased from Huineng Animal Medicine Co., Ltd (Haining, China). The heme iron obtained from the Monogastric Animal Nutrition Laboratory, Yunnan Key Laboratory of Animal Nutrition and Feed (China), was extracted from pig blood, with a purity of 98%, containing 20% heme and 2% Fe. The addition amount of three additives is the suitable addition amount of pregnant sow diets determined by previous experiments.

### 2.4. Sample Collection

On the day of birth, 10 mL and 3 mL of blood was gathered from both sows and newborn piglets from the anterior vena cava with a blood collection needle (vacuum blood collection tube without anticoagulant). After standing at 20 °C for 1 h, serum was separated by centrifugation at 3000 rpm for 15 min, then stored at −20 °C. Breast milk samples were collected on the first day, the second day, and the third day after delivery. Disinfected nipples before collection, selected 3–4 pairs of nipples before navel, and put a single nipple into a sterilized 50 mL centrifuge tube, gently squeezed the nipple root to collect breast milk in the centrifuge tube, collected 10–15 mL from each nipple, mixed the breast milk collected from each nipple evenly. Mixed the breast milk collected by each pig for three days evenly, stored at −20 °C. In total, 12 healthy newborn piglets (similar in weight, about 1.5 kg) were sacrificed in each group, syncope caused by blunt instrument hitting the head, followed by immediate bloodletting, after which heart, liver, spleen, lung, duodenum, jejunum, and ileum samples were collected within 15 min and frozen in liquid nitrogen at −80 °C for gene expression determination.

### 2.5. Analysis of Iron Content in Samples

The iron content of each sample in this experiment was determined by atomic absorption spectrometry. According to GB/T 13885-2003, the pretreatment methods of dried at 120 °C for 4 h, ash at 220 °C for 4 h, burned at 1000 °C for 2 h, and dissolved residual oxide with mixed acid were adopted. Under the conditions of analytical spectral line of 248.3 nm, slit width of 0.2 nm, flame acetylene flow rate of 1.0 L/min, and auxiliary gas air flow rate of 4.0 L/min, the iron content in it was determined by flame atomic absorption spectrometry.

### 2.6. Analysis of Serum Antioxidant Parameters

Total antioxidant capacity (T-AOC), glutathione peroxidase (GSH-Px), superoxide dismutase (T-SOD), and malondialdehyde (MDA) activities in serum were determined using relevant commercial kits (Nanjing Jiancheng Bioengineering Institute, Nanjing, China).

### 2.7. Gene Expression

Total RNA was extracted from the heart and liver using an RNA Simple Total RNA Kit (Tiangen Biotech Co., Ltd., Beijing, China). Quantitative real-time polymerase chain reaction (qRT-PCR) was performed using a Real-Time PCR Detection System (BioRad, Hercules, CA, USA) and gradient PCR instrument (Eppendorf, Hamburg, Germany). Primers were synthesized by Sangon Biotech (Shanghai) Co., Ltd. (Shanghai, China). The primers (Superoxide dismutase (*SOD*), Glutathione Peroxidase (*GSH-Px*), Thioredoxin-1 (*TRX1*), and Glutaredoxin 1 (*GRX1*) are shown in Table 2. β-actin was selected as the housekeeping gene, because its expression level was stable in all tissues of experimental pig breeds. According to the expression of β-actin gene, the expression level of mRNA was standardized, and the relative expression levels were calculated by using the 2^−∆∆Ct^ method [18].

### 2.8. Statistical Analysis

All data are presented as mean ± standard deviation (SD). In this model, four different feeds are independent variables. In the items of reproductive performance, antioxidant capacity of sows and iron content in sow tissues, sows are experimental units. Newborn piglets are experimental units in the items of iron content in tissues, serum antioxidant capacity, and antioxidant gene expression level of newborn piglets. Results were considered significant at *p* ≤ 0.05. All statistical analyses were performed using SPSS v21.0. The data were analyzed by one-way ANOVA and multiple comparisons (Duncan).

## 3. Results

### 3.1. Effects of Different Iron Additives on Reproductive Performance in Sows

According to Table 3, no remarkable differences in litter size, live litter size, or the birth weight of piglets were observed among the groups (*p* > 0.05). Nonetheless, compared with the Id group, sow litter size increased by 4.54%, 1.54%, and 6.09% in the LF, Fe-Gly, and Heme-Fe groups, respectively. Furthermore, the number of live births increased by 3.14%, 3.14%, and 1.57% in the LF, Fe-Gly, and Heme-Fe groups, respectively. Birth litter weight increased by 8.91%, 0.18%, and 0.72% in the LF, Fe-Gly, and Heme-Fe groups, respectively.

### 3.2. Effects of Different Iron Additives on Iron Content in Sow Tissue

According to Table 4, adding iron additives in diet enhanced the iron deposition in sow placenta (*p* < 0.05). Among the experimental groups, the Heme-Fe and LF groups had a higher iron content of placental (*p* < 0.001), however, there was no obvious difference between the two.

The iron content of serum in the experimental sows was higher than that in the Id group (*p* < 0.05), in the order of the LF group, the Heme-Fe group, and the Fe-Gly group. Significant differences were found among the three groups (*p* < 0.05).

Breast milk iron content was increased in the experimental groups (*p* < 0.001), in the order of the LF group, the Fe-Gly group, and the Heme-Fe group (*p* < 0.05).

### 3.3. Effects of Different Iron Additives on Iron Content in Newborn Piglet Tissue

As shown in Table 5, the serum of newborn piglets in the experimental group had the higher iron content (*p* < 0.05), with the highest content found in the Heme-Fe group (*p* < 0.001). Compared with the Id and Fe-Gly groups, serum iron content in the LF and Heme-Fe groups was increased (*p* < 0.05).

Piglets in the Heme-Fe group had the highest heart iron content (*p* < 0.001), followed by the LF group, which was obviously higher than the Id and Fe-Gly groups (*p* < 0.001). Compared with the Id group, Fe-Gly added to the pregnant sow feed also significantly increased iron content in the piglet hearts (*p* < 0.001).

Iron content in the liver, spleen, and lung of newborn piglets was higher in the LF group than in the Fe-Gly and Heme-Fe groups (*p* < 0.05). Compared with the Id group, iron content in the spleen of the Fe-Gly and Heme-Fe group piglets showed a downward trend.

Supplementation of an appropriate amount of iron in the diet of pregnant sows increased intestinal iron content in piglets, with a greater effect in the Heme-Fe group than in the other treatment groups (*p* < 0.001).

### 3.4. Effects of Different Iron Sources on Serum Antioxidant Capacity of Sows

As shown in Table 6, GSH-Px activity in sow serum was the highest in the Heme-Fe group (*p* < 0.001), the second highest level was in the LF group. (*p* < 0.001).

Serum T-AOC activity in the sows was significantly higher in the LF group than in the other three groups (*p* < 0.001), and the Id group was the lowest (*p* < 0.001). In the Fe-Gly group, serum T-AOC activity was obviously higher than that in the Id group (*p* < 0.001).

Serum T-SOD activity in the sows was the highest in the LF group (*p* < 0.001) and the second highest was the Heme-Fe group (*p* < 0.05), indicating that LF and Heme-Fe significantly increased serum T-SOD activity in the sows compared with Fe-Gly (*p* < 0.05).

The average concentration of MDA was the highest in the Id group, but there was no significant difference among the groups.

### 3.5. Effects of Different Iron Sources on Serum Antioxidant Capacity of Newborn Piglets

As seen in Table 7, the serum GSH-Px activity of the newborn piglets in the LF group was significantly higher than in the Heme-Fe and Fe-Gly groups (*p* < 0.05), however, compared with the Id group, the Fe-Gly group decreased. (*p* < 0.001).

The T-AOC activity of newborn piglets in the Fe-Gly group was the highest (*p* < 0.001). Thus, the effects of Fe-Gly on serum T-AOC activity in newborn piglets were better than those of LF and Heme-Fe.

The serum T-SOD activity in newborn piglets was the highest in the LF group (*p* < 0.05) and significantly higher in the Heme-Fe group than in Id and Fe-Gly groups (*p* < 0.05), but not significantly different between the Fe-Gly and Id groups (*p* > 0.05).

The serum MDA concentration in newborn piglets was lower in the experimental groups than in the Id group (*p* < 0.05).

### 3.6. Effects of Different Iron Sources on Antioxidant Gene Expression in Target Tissues of Newborn Piglets

Antioxidant genes *GSH-Px*, *SOD*, *GRX1*, and *TRX1* were expressed in the heart and liver of the visceral tissues of piglets, with little or no expression in the lungs and spleen.

#### 3.6.1. Analysis of Relative Expression of Antioxidant Gene mRNA in Piglet Hearts

As shown in Figure 1, *GSH-Px* expression in the piglet hearts was the highest in the LF group (*p* < 0.05), and there was no obvious difference among the other three groups. The expression of *SOD* in the LF group was the highest (*p* < 0.05). No significant differences were found among the Heme-Fe group and Fe-Gly and Id groups (*p* > 0.05). *GRX1* in the Heme-Fe group had the highest expression (*p* < 0.05), and was significantly higher in the LF and Fe-Gly groups than in the Id group (*p* < 0.05). *TRX1* expression was remarkably higher in the LF group than in the others (*p* < 0.05), showed no significant differences among the Heme-Fe group and the Fe-Gly and Id groups (*p* > 0.05), and was significantly lower in the Fe-Gly group than in the Id (*p* < 0.05).

#### 3.6.2. Analysis of Relative Expression of Antioxidant Gene mRNA in Piglet Livers

As shown in Figure 2, *GSH-Px* expression in the liver of newborn piglets was obviously higher in the LF group than in the Heme-Fe and Fe-Gly groups (*p* < 0.05), and in the Fe-Gly group it was lower than in the Id group (*p* < 0.05). Compared with the Id group, the expression of the *SOD* gene in the experimental groups was lower (*p* < 0.05), and among the Fe-supplemented groups, LF had the better results. *GRX1* in the Heme-Fe group had the highest expression (*p* < 0.05) but showed no significant difference between the Heme-Fe and LF groups (*p* >0.05). The relative expression of the *TRX1* gene was obviously higher in the LF group than in the Id, Heme-Fe, and Fe-Gly groups (*p* < 0.05).

These results indicated that dietary supplementation with LF in pregnant sows contributed to the mRNA relative expression levels of *GSH-Px* and *TRX1* genes in the liver of newborn piglets, with the best effects. The addition of iron reduced the expression of *SOD*, while administration of LF and Heme-Fe increased the expression of *GRX1*.

## 4. Discussion

The bioavailability of iron from different sources is not the same [19]. According to previous experiments, we selected iron additives suitable for this study and their dosage, and they can provide similar iron concentrations for sows [20].

### 4.1. Effects of Different Iron Sources on Reproductive Performance in Sows

Late pregnancy is an important period for the fetal digestive system and muscle development, with increases in fetal weight mainly affected by maternal dietary protein and energy levels [17]. The nutritional requirements of sows and their offspring increase in the course of the second half of pregnancy [21]. An insufficient supply of nutrients during this period can lead to stunted fetal growth and low birth weight [22]. Iron reserves in sows decrease with increasing parity, and older multiparity sows are prone to iron deficiency [23]. Severe iron deficiency can impair reproductive performance in sows, while iron supplementation in the course of the second and third trimesters during peak fetal growth increases piglet weight [24].

In the current study, we provided sows with experimental diets 80 days before parturition. The results showed that dietary supplementation with LF, heme iron, and Fe-Gly during mid to late gestation had no significant effect on litter size, live litter size, and litter weight at birth, which was different from previous reports [25,26]. Tummaruk et al. showed that the addition of glycine chelates to sow diets during late pregnancy can improve the growth rate of suckling piglets and reduce the mortality of newborn piglets [24]. In contrast, Wei et al. reported that iron supplementation in sows has no remarkable effect on total weight or average weight of live-born and weaned piglets [27]. Furthermore, Bhattarai et al. reported that intramuscular iron injections (2500 mg every 2 weeks) in the second trimester do not reduce stillbirths [28] and Liu et al. reported that LF supplementation does not affect the reproductive performance of sows [29]. Thus, these results suggest that the addition of iron in different periods and in different individual animals may lead to differences in experimental results.

### 4.2. Effects of Different Iron Additives on Iron Content in Sow and Neonate Piglet Tissues

Iron deficiency is common in piglets in the early postnatal period, and the most serious case will lead piglets to develop iron deficiency anemia (IDA), which is prevalent in early postnatal piglets [30]. In the course of gestation, insufficient iron supply may lead to decreased neonatal iron status and maternal iron deficiency anemia [31]. However, the research on the molecular regulation of iron absorption in newborn piglets is rare. Several studies have concluded a limited effect of adding dietary iron supplements directly to piglet diets on preventing iron deficiency anemia [32,33], whereas Dong et al. found that the oral administration of Fe-Gly can improve oxidative and iron homeostasis [8]. Iron in newborn piglets mainly comes from the mother. Previous research has reported that the addition of Fe-Gly to sow diets can significantly increase iron content in the placenta and increase iron saturation and decrease piglets’ total iron-binding capacity at birth [34]. Furthermore, the replacement of FeSO4 supplements with Fe-Gly in late gestation can increase litter birth weight, likely via enhanced iron transport in the placenta [34]. The supplementation of LF in sow diets can also improve pig production performance, milk production, serum immunoglobulin A (IgA), and secretory IgA (SIgA) levels [26]. Heme iron, as an iron supplement to treat human iron deficiency, is rarely used to treat pregnant sows at present [35]. Medical research shows that for pregnant women, heme iron is a dietary iron source with higher bioavailability than FeSO_4_. [36] Heme iron used in this study was extracted from pig blood by our research group, which showed good bioavailability in previous experiments [37].

In this study, supplementation with heme iron, glycine iron, and LF significantly increased placental iron and serum iron in sows and serum iron in suckling piglets. Compared with the Id group, the placental iron content of sows in the treatment group was higher, and there was no significant difference between the LF group and Heme-Fe group, but the situation in the tissues of newborn piglets was different, which may be caused by the influence of the placental barrier. Considering the serum iron content of newborn piglets, LF and Heme-Fe showed a stronger ability to cross the placental barrier and different iron additives had different iron deposition sites in newborn piglets. The iron deposition in the lung, spleen, and liver of the lactoferrin group was much higher than that of the Heme-Fe and Fe-Gly. However, in the intestine, the iron deposition in the Heme-Fe group was the most, which was related to the serum iron content of newborn piglets, which proved that LF and Heme-Fe had different effects on the iron content of newborn piglets, which may be related to their own properties and needs further study. Compared with LF and Heme-Fe, Fe-Gly had no excellent ability to cross the placental barrier, and its deposition in tissues was not prominent. The treatment groups also exhibited an increase in milk iron content, with LF showing the most obvious effects. This means that sows in the treatment group can provide more iron to suckling piglets through breastfeeding. Compared with the Id group, the increase in the LF group was as high as 138.4%, and that in heme iron group and glycine iron group was 32.7% and 63.5%, respectively. Although the final effect on suckling piglets is affected by the absorptive capacity of piglets’ intestines, the increase in iron content in milk is also meaningful. However, further research is needed with suckling piglets.

### 4.3. Effects of Different Iron Sources on Antioxidant Capacity of Sows and Newborn Piglets

Due to its unique chemical properties, iron affects redox reactions in the body. Iron has the ability to transform between different oxidation states, which enables it to transfer electrons between organic compounds and react in oxidation reactions, which is important for single-electron transfers to induce free-radical processes [38]. Under normal circumstances, the production and clearance of free radicals in the body are in a state of dynamic balance, whereby free radicals are constantly produced and eliminated by the antioxidant system. Under conditions such as oxidation and environmental stress, the balance of oxygen free radicals in the body is destroyed, cellular free radicals increase, and the antioxidant capacity of tissues decreases, resulting in excessive free radicals [39]. For example, during the phagocytosis of leukocytes, superoxide anions (O^2−^) and hydrogen peroxide (H_2_O_2_) are produced due to increased oxygen consumption [40]. Under the action of free iron ions, active hydroxyl radicals (•OH) are easily formed, thus causing oxidative damage to the body [41].

LF plays a considerable part in the regulation of cellular redox activity, protects cells against oxidative damage, and improves the antioxidant capacity of the body. Safaeian et al. reported that LF pretreatment can reduce H_2_O_2_ levels and increase intracellular and extracellular iron-reducing antioxidant capacities [42]. Buey et al. reported that LF can regulate oxidative stress and inflammatory responses caused by the activation of Toll-like receptors (TLR) [43]. Jia et al. showed that LF has a protective effect on damaged intestinal epithelial cells and can maintain cell viability [44]. Park et al. showed that LF protects neurons against neurotoxicity mediated by prion protein (PrP) (106–126) by scavenging intracellular reactive oxygen species (ROS) [45]. Zalutski et al. demonstrated that LF disrupts antioxidant balance by blocking the cell cycle and malignant cell apoptosis in the G2/M phase by increasing ROS levels, free iron, and NO production rates [46]. Liu et al. showed that LF can alleviate the symptoms of Parkinson’s mice and enhance the ability of resisting iron imbalance and antioxidant stress [47]. These in vitro experiments proved that lactoferrin can improve the anti-oxidation and anti-aging ability of cells and reduced the degree of oxidative stress and inflammatory reaction in various ways.

Supplementing LF in piglets can increase SOD and T-AOC activities in serum [48] and reduce MDA content in serum and longissimus muscle [49], as also found in mice [50]. An et al. showed that the addition of LF in piglet diets not only changes biochemical parameters (T-AOC, SOD, GSH-Px, MDA) in serum, but also the expression levels of antioxidant genes (*GRX1*, *SOD*, *GSH-Px*) in the heart and liver [51]. Kruzel et al. reported that LF not only regulates intracellular ROS levels, but also up-regulates the expression of *GPX* and *SOD*. Kruzel’s research showed that it is feasible to improve the oxidative stress ability of cells by increasing gene expression, and the high expression of antioxidant genes can improve anti-inflammatory ability [52]. The results of our experiment showed that dietary LF supplementation increased serum antioxidant parameters in pregnant sows and newborn piglets, indicating that maternal LF supplementation had beneficial effects on the antioxidant capacity of both. However, during pregnancy, sows and piglets were connected by umbilical cords, so it cannot be clearly said that the changes of serum antioxidant biochemical indexes of piglets were caused by the changes of their own gene expression level. In fact, we thought that for newborn piglets, the internal environment of piglets was more affected by maternal blood circulation. Dietary LF increased the expression of antioxidant genes in the heart of piglets but decreased the *SOD* expression level in the liver. In fact, the *SOD* expression level in the liver of the treatment group decreased to varying degrees, but it increased to varying degrees in the heart, which may be caused by the change of SOD level in the blood, the maintenance of balance in the body, or that the use of iron additives caused a certain load on the liver of newborn piglets, but no liver diseases were observed in piglets in the same litter (not slaughtered, raised on the farm). These results indicate that LF supplementation in pregnant sows can promote the antioxidant capacity of newborn piglets, but the effect on the liver of newborn piglets needs to be further studied.

Both Heme-Fe and non-Heme-Fe are oxidation catalysts in muscle tissue, and certain heme iron oxidation reactions are thought to be explained by a Fenton-like mechanism [53]. Notably, hydroxyl radicals are formed in the reaction, or LO• in the case of LOOH reduction [54]. Heme-Fe reacts with peroxides to produce free radicals, which affect the organism’s oxidative environment. Decomposition of Heme-Fe is accomplished by the clearance of aged and damaged red blood cells by tissue macrophages, especially in the spleen [55,56]. The breakdown of heme releases iron, which increases oxidative stress [57]. Tang et al. reported that heme iron-enriched peptides can significantly relieve iron deficient anemia (IDA), with strong antioxidant activity in vitro and in vivo [58]. Our results showed that dietary supplementation with Heme-Fe increased serum antioxidant parameters in pregnant sows and newborn piglets, suggesting beneficial effects on the antioxidant capacity of sows. Serum GSH-Px activity in newborn piglets was significantly reduced and *SOD* and *GRX1* expression levels were increased in the heart and liver of newborn piglets, indicating that maternal supplementation of Heme-Fe had obvious effects on piglets, with improved expression of antioxidant-related genes but decreased biochemical activity in the serum. However, compared with the other two treatment groups, the serum MDA content of newborn piglets in the Heme-Fe group was higher, which indicates that Heme-Fe may have a higher degree of oxidative damage. The specific reasons need further study.

Fe-Gly is an organic iron, and a common iron supplement in piglet production. Ma et al. reported that dietary supplementation of Fe-Gly improves iron tissue storage, growth performance, and antioxidant status in broilers [59]. Feng et al. reported that Fe-Gly addition can have a linear effect on CAT and succinate dehydrogenase (SDH) activities and increase the SOD and SDH activities of pigs’ livers [60]. Rao et al. reported that SOD activity decreases when rats are fed iron-deficient diets [61]. Our study showed that dietary Fe-Gly supplementation increased the serum antioxidant parameters of pregnant sows, thereby demonstrating a favorable effect on the antioxidant capacity of sows. Similar to the Heme-Fe group, however, piglets showed significantly decreased GSH-Px activity in serum and increased *SOD* and *GRX1* expression in the heart. The expression of antioxidant genes in the liver of the newborn piglets in the Fe-Gly group decreased more markedly.

The effects of three iron additives on the antioxidant genes’ expression level were compared and dietary supplementation in sows with LF, Heme-Fe, and Fe-Gly contributed to the relative expression of *GRX1* in the hearts of newborn piglets. LF contributed more to the relative expression levels of *GSH-Px*, *SOD*, and *TRX1* of newborn piglets in the hearts compared with dietary supplementation with Heme-Fe and Fe-Gly.

## 5. Conclusions

In summary, dietary LF, Heme-Fe, and Fe-Gly improved the antioxidant capacity of sows during mid to late gestation, but they have no significant effect on reproductive performance. All three iron additives can increase the iron level in the placenta, serum, and breast milk of sows, and LF can increase the iron level in serum and breast milk the most. Moreover, the iron deposition caused by different iron agents was different in newborn piglets. LF can increase the iron level in the lung, spleen, and liver the most, and Heme-Fe can increase the iron content in the intestine the most. The supplementation of LF increased the antioxidant capacity and antioxidant gene expression in piglet serum. Heme-Fe increased the expression level of genes but decreased the level of serum biochemical indices. Dietary supplementation with Fe-Gly had no significant effect on the antioxidant capacity of piglets, while the expression of many antioxidant genes in the heart and liver decreased significantly. Therefore, in this study, LF appears to be a better maternal iron supplement.

## Figures and Tables

**Figure 1 animals-13-00517-f001:**
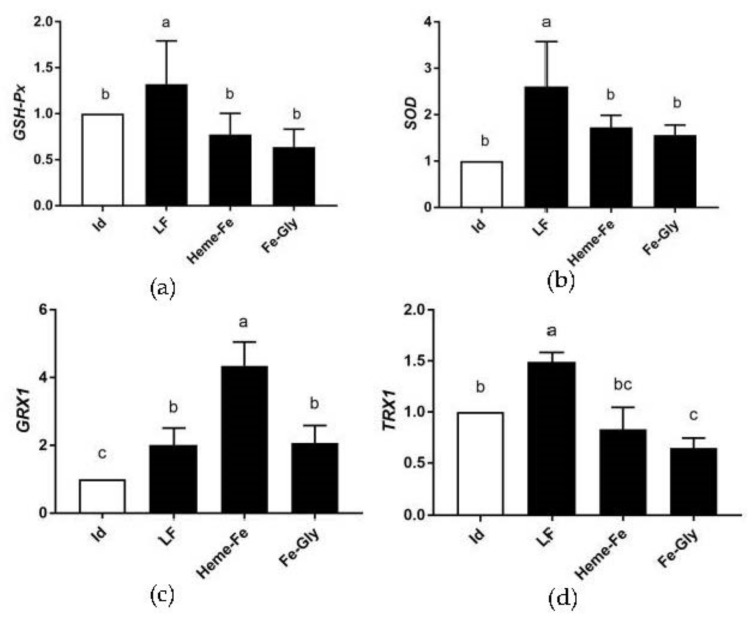
Effects of different iron sources on *GSH-Px* (**a**), *SOD* (**b**), *GRX1* (**c**), and *TRX1* (**d**) gene expression in neonate piglet heart. Id: iron deficiency group; LF: lactoferrin group; Heme-Fe: heme iron group; Fe-Gly: Fe glycinate group. Values in columns with different small letters indicate significant difference (*p* < 0.05).

**Figure 2 animals-13-00517-f002:**
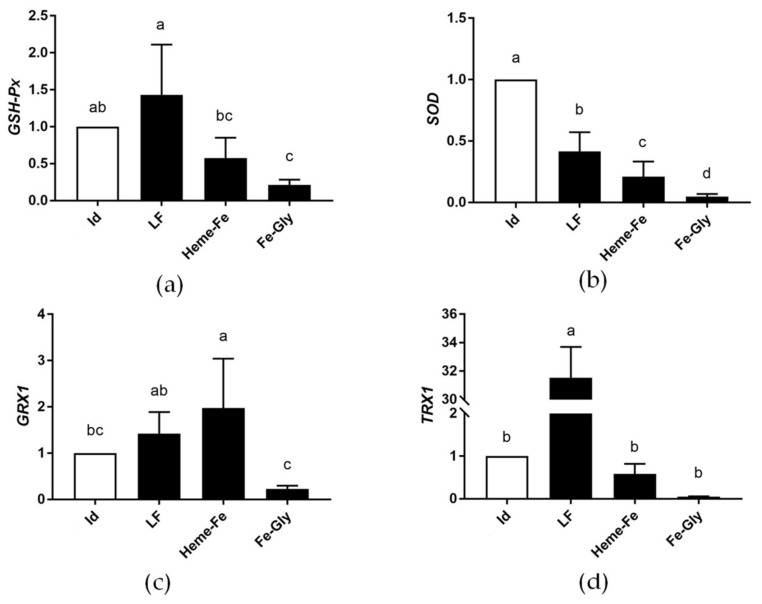
Effects of different iron sources on *GSH-Px* (**a**), *SOD* (**b**), *GRX1* (**c**), and *TRX1* (**d**) gene expression in neonate piglet liver. Id: iron deficiency group; LF: lactoferrin group; Heme-Fe: heme iron group; Fe-Gly: Fe glycinate group. Values in columns with different small letters indicate significant difference (*p* < 0.05).

**Table 1 animals-13-00517-t001:** Composition and nutrient levels of basal diet (air-dry basic).

Ingredient	Content (%)	Nutrient Level ^2^	
Corn	67.00	DE (MJ/kg)	13.10
Soybean meal	18.50	CP	15.05
Wheat bran	7.50	Lys	0.90
Fish meal	1.00	Met	0.23
Soybean oil	2	Thr	0.60
Premix ^1^	4.00	Trp	0.17
Total	100	Ca	0.75
		AP	0.30
		Fe (mg/kg)	80.09

^1^ Premix provides (per kg): VA 8000 IU; VD3 1200 IU; VE 60 IU; VK3 2 mg; VB1 1 mg; VB2 4 mg; VB6 1 mg; VB12 17.00 μg; folic acid 1.3 mg; biotin 0.2 mg; pantothenic acid 12 mg; choline 1 g; Cu 12 mg, Mn 25 mg; Zn 100 mg; Se 0.30 mg; I 0.4 mg. ^2^ DE was calculated, others were measured.

**Table 2 animals-13-00517-t002:** Information on primers used for RT-PCR.

Genes	Primer Sequence (5′-3′)	Primer Length/nt	Amplification Length/bp	GenBank Accession No.
*SOD*	F:CAGGTCCTCACTTCAATC	18	254	NM_001190422.1
	R:CAAACGACTTCCAGCAT	17		
*GSH-Px*	F:AGAAGTGTGAGGTGAATGGC	20	325	NM_214201.1
	R:CCCGAGAGTAGCACTGTAAC	20		
*GRX1*	F:TTTTCATCAAGCCCACC	17	196	NM_214233.1
	R:CCACCTATACACTCTTTACCG	21		
*TRX1*	F:CAAGCCTTTCTTCCATTC	18	148	NM_214313.2
	R:ACCCACCTTCTGTCCCT	17		
*β*-actin	F:TCTGGCACCACACCTTCT	18	114	NM_214313.2
	R:TGATCTGGGTCATCTTCTCAC	21		

**Table 3 animals-13-00517-t003:** Effects of iron additives on reproductive performance in sows.

Item ^1^	Id Group	LF Group	Heme-Fe Group	Fe-Gly Group	*p*-Value
Litter size (n)	11.00 ± 1.41	11.50 ± 1.87	11.67 ± 1.75	11.17 ± 1.72	0.900
Size of piglets born alive (child/only)	10.83 ± 1.72	11.17 ± 1.72	11.00 ± 1.41	11.17 ± 1.33	0.979
Litter weight at birth (kg)	16.61 ± 0.94	18.09 ± 2.47	16.73 ± 2.14	16.64 ± 1.14	0.430

^1^ Results are presented as mean ± SD.

**Table 4 animals-13-00517-t004:** Effects of different iron additives on iron content in sow.

Item ^1^	Id Group	LF Group	Heme-Fe Group	Fe-Gly Group	*p*-Value
Placenta (μg/g)	20.55 ± 0.19 ^c^	34.74 ± 0.20 ^a^	34.84 ± 0.19 ^a^	31.10 ± 0.14 ^b^	<0.001
Serum (mg/L)	19.40 ± 0.12 ^d^	31.80 ± 0.26 ^a^	25.11 ± 0.25 ^b^	23.91 ± 0.13 ^c^	<0.001
Breast milk (mg/L)	1.56 ± 0.01 ^d^	3.72 ± 0.02 ^a^	2.07 ± 0.01 ^c^	2.55 ± 0.02 ^b^	<0.001

^1^ Results are presented as mean ± SD. ^a, b, c, d^ Within a row, values with different letter superscripts differ significantly (*p* < 0.05).

**Table 5 animals-13-00517-t005:** Effects of different iron additives on iron content in newborn piglet tissue.

Item ^1^	Id Group	LF Group	Heme-Fe Group	Fe-Gly Group	*p*-Value
Heart (μg/g)	20.89 ± 0.01 ^d^	30.62 ± 0.14 ^b^	33.42 ± 0.085 ^a^	23.27 ± 0.20 ^c^	<0.001
Liver (μg/g)	26.38 ± 0.18 ^d^	67.76 ± 0.41 ^a^	43.01 ± 0.18 ^b^	40.85 ± 0.27 ^c^	<0.001
Spleen (μg/g)	49.03 ± 0.07 ^b^	67.05 ± 0.38 ^a^	47.26 ± 0.35 ^c^	45.85 ± 0.51 ^c^	<0.001
Lung (μg/g)	24.64 ± 0.12 ^c^	32.43 ± 0.44 ^a^	27.73 ± 0.47 ^b^	24.55 ± 0.28 ^c^	<0.001
Duodenum (μg/g)	19.43 ± 0.10 ^d^	25.75 ± 0.18 ^b^	30.81 ± 0.06 ^a^	22.90 ± 0.35 ^c^	<0.001
Jejunum (μg/g)	18.26 ± 0.08 ^d^	28.42 ± 0.28 ^b^	32.19 ± 0.08 ^a^	21.23 ± 0.04 ^c^	<0.001
Ileum (μg/g)	19.60 ± 0.21 ^d^	24.81 ± 0.18 ^b^	31.43 ± 0.20 ^a^	22.48 ± 0.17 ^c^	<0.001
Serum (mg/L)	16.47 ± 0.19 ^d^	34.69 ± 0.12 ^b^	37.2 ± 0.39 ^a^	25.53 ± 0.07 ^c^	<0.001

^1^ Results are presented as mean ± SD. ^a, b, c, d^ Within a row, values with different letter superscripts differ significantly (*p* < 0.05).

**Table 6 animals-13-00517-t006:** Effects of iron sources on serum antioxidant indices in sows.

Item ^1^	Id Group	LF Group	Heme-Fe Group	Fe-Gly Group	*p*-Value
GSH-Px/(U/mL)	133.65 ± 4.16 ^d^	163.58 ± 5.01 ^b^	178.66 ± 5.95 ^a^	154.54 ± 5.42 ^c^	<0.001
T-AOC/(U/mL)	6.25 ± 0.43 ^d^	14.76 ± 0.39 ^a^	10.59 ± 0.18 ^b^	8.97 ± 0.28 ^c^	<0.001
T-SOD/(U/mL)	56.73 ± 1.07 ^c^	78.07 ± 2.65 ^a^	62.11 ± 1.53 ^b^	57.44 ± 1.04 ^c^	<0.001
MDA/(nmol/mL)	6.74 ± 0.45	6.54 ± 0.50	6.61 ± 0.10	6.38 ± 0.17	0.663

^1^ Results are presented as mean ± SD. ^a, b, c, d^ Within a row, values with different letter superscripts differ significantly (*p* < 0.05).

**Table 7 animals-13-00517-t007:** Effects of iron sources on serum antioxidant indices in neonate piglets.

Item ^1^	Id Group	LF Group	Heme-Fe Group	Fe-Gly Group	*p*-Value
GSH-Px/(U/mL)	130.80 ± 11.39 ^ab^	133.62 ± 7.42 ^a^	121.66 ± 3.64 ^bc^	113.78 ± 6.70 ^c^	<0.001
T-AOC/(U/mL)	5.26 ± 0.12 ^d^	5.59 ± 0.16 ^c^	10.56 ± 0.28 ^b^	11.66 ± 0.19 ^a^	<0.001
T-SOD/(U/mL)	46.48 ± 1.79 ^c^	53.88 ± 1.86 ^a^	49.41 ± 2.89 ^b^	45.61 ± 1.96 ^c^	<0.001
MDA/(nmol/mL)	10.56 ± 0.49 ^a^	6.49 ± 0.29 ^c^	9.18 ± 0.22 ^b^	6.99 ± 0.69 ^c^	<0.001

^1^ Results are presented as mean ± SD. ^a, b, c, d^ Within a row, values with different letter superscripts differ significantly (*p* < 0.05).

## Data Availability

Data can be accessed in the article.

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
