# Peer review of "Effects of Different Iron Supplements on Reproductive Performance and Antioxidant Capacity of Pregnant Sows as Well as Iron Content and Antioxidant Gene Expression in Newborn Piglets"

_animals, 2023, doi:10.3390/ani13030517_

Round 1

Reviewer 1 Report

Dear authors,

Please, find all specific comments, suggestions and questions in the enclosed file.

An English review would benefit the manuscript. Overall, this is an interesting manuscript. There are some precisions to be made at the methodology section and some precisions/corrections to be made at the results section. The big weakness of this manuscript is the discussion section where the authors basically repeated the results section and cited many studies from others authors that agree or not with their own results. There is no real explanation of the results. It is fine to state that iron affects the immune system but the authors did not explore the metabolism of iron trying explain how it happens. That is what will bring new knowledge on the matter or at least inspire others to investigate further.

Thank you

Author Response

Animals

Original Manuscript ID: animals- 2136292

Title: Effects of different iron supplements on reproductive performance and antioxidant capacity of pregnant sows as well as iron content and antioxidant gene expression in newborn piglets General comments

An English review would benefit the manuscript. Overall, this is an interesting manuscript. There are some precisions to be made at the methodology section and some precisions/corrections to be made at the results section. The big weakness of this manuscript is the discussion section where the authors basically repeated the results section and cited many studies from others authors that agree or not with their own results. There is no real explanation of the results. It is fine to state that iron do this or that with the immune system but the authors did not explore the metabolism of iron trying explain how it happens.

That is what will bring new knowledge on the matter or at least inspire others to investigate further.

Specific comments

Abstract; L16-18: How to prevent anemia in piglets by studying antioxidant capacity? Please, re-word.

Re: Line 16, changed to “To improve the reproductive performance of sows and the iron nutrition of newborn piglets,”

Abstract; L18: The authors describe the sows. What about their respective litters? They should be described as well. They were used from birth to weaning?

Re: This experiment did not involve suckling piglets, and newborn piglets were sampled after birth. Unsampled sows and piglets continue to be raised on the farm.

Abstract; L18-21: Here there are only 3 groups. Is the control group missing?

Re: Line 19, added “the control group was fed basic diet,”

L23: Piglets. Please, be more specific on age (at birth, at weaning?)

Re: Line 26, changed to “newborn piglets”

L24-25: Please, give some directions (overall, what the expression of these genes mean?) readers can better understand what kind of effects were observed.

Re: Line 24, changed to “3) The addition of different iron sources improved the level of serum antioxidant biochemical indexes of sows and newborn piglets, and it can have an effect on gene level, among which lactoferrin has the best effect.”

L26: Third trimester of pregnancy. It should be mentioned earlier in the abstract to better describe what kind of animals were used. In fact, in the methodology it is stated that sows were supplemented during 80 days before farrowing, which means from day 35 of gestation. Therefore, sows were supplemented during the second and third trimester of gestation.

Re: Line 29, changed to “during the second and third trimester of gestation”

L27-28: In the simple summary it was stated that " lactoferrin and heme iron can improve the iron nutrition". Why it was not added in the abstract?

Re: Line 31, added “and the iron nutrition”

L37-38: Please, add a reference.

Re: Line 42, references have been added.

L40: maternal inorganic iron supplementation.

Re: Line 44, changed to “ferrous sulfate”

L52: LF. Please, define at first use.

Re: Line 56, definitions have been added

L61: Fluid circulation. Would it be “blood circulation”?

Re: Line 65, changed to “blood circulation”

L67 and 68: Consumed. Would it be “absorbed”?

Re: Line 71 and 72, changed to “absorbed”

L92: Please, add information of average body weight.

Re: Line 96, added “similar in weight”

L96: Considering that farrowing may vary from 113- to 116 days, I believe the authors have established a fixed day to start supplementation (for example 35 days of gestation). Please, indicate exactly when supplementation started and ended.

Re: Line 99, added “From the 33rd day of pregnancy until delivery,”

L98-99: According to the NRC (2012), gestating and lactating sows should be supplemented with 80 mg/kg of iron (independently of iron levels from feed ingredients). Therefore, the basal diet without iron supplementation is not in accordance with the NRC (2012).

L100: I respect the authors decision on using a control diet without supplemental iron but I believe it would make more sense (or at least would give more practical information) to use inorganic Fe at levels recommended by NRC or the Chinese standards.

L102: Why such different levels of iron between diets (for example: 200 vs 500 mg/kg)? Are these actual Fe levels or are they product levels? Do these levels provide similar amounts of elemental iron? It should be explained somewhere.

Re: Added description on line 114.

The feed formula of this experiment refers to NRC standard and China pig raising standard, and the basic diet contains 80mg/kg of iron. In our previous study, ferrous sulfate, an inorganic iron additive, was used, but it had little effect on newborn piglets. This experiment mainly wanted to compare the effects of two kinds of organic iron and heme iron additives. The control group was established to verify the effects of these three iron additives, and no additional iron was added, so it can also be called iron deficiency group. What is mentioned in the article is the product content (for example, 200mg/kg), which is because sows have different utilization rates of different additives, and it is not appropriate to only consider the consistency of iron content. The addition amount of the three additives is the suitable addition amount of pregnant sows determined by our previous experiments. Heme iron is provided by the Single Stomach Animal Nutrition Laboratory of Yunnan Key Laboratory of Animal Nutrition and Feed, and extracted from pig blood. The purity is 98%, the heme content is 20%, the iron content is 2%, and the remaining components are HGB, peptides, amino acids and so on.

L114-115: It gives the impression that 3 ml was collected from sows and 10 ml from piglets, what I believe was not the case. Please, re-word for better understanding.

Re: Line 121, changed to “On the day of birth, 10 mL and 3 mL of blood was gathered from both sows and newborn piglets from the anterior vena cava with a blood collection needle (vacuum blood collection tube without anticoagulant).”

L115: Please, describe collection tubes characteristics (trace-element free tubes? any preserving agent?)

Re: Line 122, added “(vacuum blood collection tube without anticoagulant)”

L117: How many piglets from each sow? Were them equally distributed between sows?

Re: The size of litters per sow is different (ranging from 9 to 14). Newborn piglets with similar weight were selected in the experiment. Line 127, added “(similar in weight, about 1.5kg)“”

L130: Please, describe how the housekeeping gene was selected.

Table 2: Please, describe genes in the footnote.

Re: Line 149, changed to “The primers (Superoxide dismutase (SOD), Glutathione Peroxidase (GSH-Px), Thiore-doxin-1 (TRX1) and Glutaredoxin 1 (GRX1) are shown in Table 2. β-actin was selected as the housekeeping gene, because its expression level was stable in all tissues of experimental pig breeds. According to the expression of β -actin gene, the expression level of mRNA was standardized, and the relative expression levels were calculated by using the 2−∆∆Ct method [56].”

L132: Please, indicate independent variables and experimental unit.

Re: Line 158, added “In this model, four different feeds are independent variables. In the items of repro-ductive performance, antioxidant capacity of sows and iron content in sow tissues, sows are experimental units. Newborn piglets are experimental units in the items of iron content in tissues, serum antioxidant capacity and antioxidant gene expression level of newborn piglets.”

L139: Litter number. Would it be “litter size”?

Re: Line 167, changed to “litter size” and “live litter size”

L140: Birth weight of sows. Would not it be “birth weight of piglets”?

Re: Line 168, changed to “birth weight of piglets”

L148: Tables should stand alone, so this footnote should be repeated in all tables.

Table 4: Please, indicate the units for each item. Same for Table 3 (although more obvious) and Table 5.

Re: Footnotes have been added after each table, and the units in tables 3 and 5 have been indicated.

L161: Piglets. It should state “newborn piglets”.

Re: Line 191, changed to “newbron piglets”

L171: Spleen. Not only Fe-Gly but also Heme-Fe.

L171: Lung. This was the case only for heme-Fe since Fe-Gly was not different from Control.

Re: Line 200, changed to “Compared to the control group, iron content in the spleen of Fe-Gly and Heme-Fe group piglets showed a downward trend.”

L178-180: This sentence is just repeating the previous one.

Re: This sentence has been deleted.

L185-186: Repeating previous sentences.

Re: This sentence has been deleted.

L190-191: This sentence may lead readers to a misunderstanding. Please, clearly state that not statistical differences were observed among groups.

Re: Line 223, changed to “The average concentration of MDA was the highest in the control group, but there was no significant difference among the groups.”

L191-192: The authors cannot claim a tendency for reduced MDA in Fe-supplemented animals because, although numerical values were different, P values were far from a tendency.

Re: This sentence has been deleted.

L195-196: This is incorrect. Control group was not different from LF group.

Re: Line 232, changed to “As seen in Table 7, serum GSH-Px activity in the newborn piglets was the highest in the LF group, significantly higher than Heme-Fe and Fe-Gly groups (p < 0.05), however, compared with the control group, the Fe-Gly group de-creased. (p < 0.001).”

L197-199: This sentence is just repeating the previous one.

Re: This sentence has been deleted.

L205-207: This sentence is just repeating the previous one.

Re: This sentence has been deleted.

L209: LF and Fe-Gly were not different among them.

Re: Line 245, changed to “The serum MDA concentration in newborn piglets was lower in the experimental groups than in the control group (p < 0.05), the lowest was the LF group.”

L209-211: This sentence is just repeating the previous one.

Re: This sentence has been deleted.

L216: According to the NCBI, these genes were supposed to have a fair expression in these two tissues. Do the authors have any explanation for the absence of expression?

Re: Line 153, added “According to the expression of β -actin gene, the expression level of mRNA was standardized, and the relative expression levels were calculated by using the 2−∆∆Ct method [56].”

L221: During. Would not it be “between”?

L224-225: Please, re-word. It gives the impression that control, Heme-Fe and Fe-Gly were not different among them.

Re: Line 259, changed to “No significant differences were found among the Heme-Fe group and Fe-Gly and control groups (p > 0.05).”

L227-231: This sentence should be moved to the discussion section.

Re: Modified and transferred to line 453.

L234: LF. Please, define.

Re: LF group is the name of the group to which lactoferrin is added, which has been defined in the abstract and section 2.3.

Figure 1: “Same below” and “Same for figure 2”. Figures should stand alone, so this footnote should be repeated in all figures.

Re: The illustration has been supplemented and improved.

L239: Should state that it was not different from control.

Re: Line 278, changed to “and in the Fe-Gly group was lower than in the control group”

L242: It is not appropriate. LF expression of half of that of the control group. I believe the authors meant that among the Fe-supplemented groups, LF had the better results.

Re: Line 281, changed to “among the Fe-supplemented groups, LF had the better results.”

L258: Supply. It should be “supply of nutrients”

Re: Line 300, changed to “supply of nutrients”

L266: Late gestation. It should be “mid to late gestation”.

Re: Line 308, changed to “mid to late gestation”

L266-267: According to statistics, there was no effect. The "n" necessary to find differences for these parameters can be discussed but in the present experimental conditions, the results express no difference between treatments.

L279: There is a contradiction here. I believe the authors should not try to sell that iron is good for reproductive performance because their own results say it had no positive effects. I was expecting the authors to explain, based on metabolism and physiology, why it happened.

Re: This discussion has been revised. “In the current study, we provided sows with experimental diets 80 days before parturition. Results showed that dietary supplementation with LF, heme iron, and Fe-Gly during mid to late gestation had no significant effect litter size, live litter size, and litter weight at birth, which was different from previous reports [21, 22]. Tummaruk et al. showed that the addition of glycine chelates to sow diets during late pregnancy can improve the growth rate of suckling piglets and reduce mortality of newborn piglets [20]. In contrast, Wei et al. reported that iron supplementation in sows has no remarkable effect on total weight or average weight of live-born and weaned piglets [23]. Fur-thermore, Bhattarai et al. reported that intramuscular iron injections (2 500 mg every 2 weeks) in the second trimester do not reduce stillbirths [24] and Liu et al. reported that LF supplementation does not affect the reproductive performance of sows [25]. Thus, these results suggest that the addition of iron in different periods and in different in-dividual animals may lead to differences in experimental results.”

L295: What did the authors mean by “improve pig production”?

Re: Line 338, changed to “production performance”

L297-299: It is only repeating the results section. Please, present an explanation and/or hypothesis for that.

L299-301: This is a very interesting result. Please, present an explanation and/or hypothesis for that. In the introduction it was stated that under regular Fe supplementation strategies, milk Fe cover approximately 10-20% of piglets requirements. The authors could come up with similar calculation based on these results since this could have practical implication in terms of IM Fe supplementation to piglets.

L301-304: Again, only repeating results section. Please, present an explanation and/or hypothesis.

L304-305: It is only repeating parts of the previous statement.

Re: L341-L361, according to your suggestion, this discussion has been rewritten.

Item 4.3: Instead of discussing each treatment individually, the authors could have made an overall discussion comparing sows and piglets antioxidant system. Then they could have discussed the possible explanations for the results (better bioavailability of some sources, eventual different metabolic pathways between sources, etc)

L321-332: Please, indicate if those studies were in vitro or in vivo. Additionally, please present an explanation and/or hypothesis.

Re: Line 393, added “These in vitro experiments proved that lactoferrin can improve the anti-oxidation and anti-aging ability of cells and reduced the degree of oxidative stress and inflammatory reaction in various ways.”

L341-344: Repeating results section. Please, explain results.

L345: Liver of piglets vs SOD in liver. There is a contradiction here.

L346-348: This overall conclusion is fine, but the authors must try to explain how it happens.

Re: L407-L418, according to your suggestion, this discussion has been rewritten.

L356: Please, define abbreviation.

Re: Line 428, added “iron deficient anemia (IDA)”

L364: It is fine to state that if there are not many information on the matter but the authors could try to come up with some hypothesis that would guide further studies.

Re: Line 436, changed to “However, compared with the other two treatment groups, the serum MDA content of newborn piglets in Heme-Fe group is higher, which indicates that Heme-Fe may have a higher degree of oxidative damage. The specific reasons need further study.”

L378-382: It should be moved to the conclusion section and summarized.

Re: This part has been combined with the conclusion.

L384-385: Improve reproductive performance. It is incorrect according to the present study results.

L388: What did the authors mean by “surface level”?

L388-389: What did the authors mean by “decreased the activity of serum biochemical indices”?

L389-391: Not a conclusion but a summary of results.

L391: It repeats L382.

Re: L465-L476, according to your suggestion, the conclusion has been rewritten.

Reviewer 2 Report

Few studies have compared the effects of different iron supplements on pregnant sows and resulting piglets. This manuscript applied three different kinds of iron sources (heme iron, organic iron, and LF) in sow production and newborn metabolism. It highlights iron supplementation could have a positive effect on reproductive performance in sows. Interestingly, supplementation of LF in pregnant sow better additive effects on the antioxidant capacity than the others. In general, the study revealed important choices of iron source and potentially beneficial role of iron for the metabolism of sow and offspring.

1. In this experiment, why do you choose heart and liver as the expression tissues of antioxidant genes?

2. Are the contents of iron in the diets of the three treatment groups the same? How to determine whether the difference is caused by the kind of iron additive instead of the iron content? Please supplement the composition of lactoferrin used.

3. Please supplement the specific determination methods of iron content in various tissues and serum.

4. Discussion 4.2: Please analyze the relationship between piglets and sows. The discussion part failed to connect the relationship between the iron content of piglets and sows.  Refer to: Dan Wan, et al. Maternal dietary supplementation with ferrous N-carbamylglycinate chelate affects sow reproductive performance and iron status of neonatal piglets. Animal, 2018

L22: The three additives improved iron nutrition in piglets, with LF and Heme-Fe showing stronger effects. What do you mean? stronger effects on what?

Line 42: “an analogous” instead of “a analogous”.

Line 42: “a necessary” instead of “an necessary”.

Line 54: “a high” instead of “high”.

Line 63: “disorders” instead of “disorder”.

10. Line 142: “in the LF, Fe-Gly, and Heme-Fe” instead of “in the LF, Fe-Gly, Heme-Fe”.

L117: how did you choose the sacrificed newborn piglets in each group? According to body weight? describe it in detail.

L118: In sample collection part, the collection of breast milk, duodenum, jejunum and ileum is missing.

L131: what is GRX1 and TRX1? Please add the gene annotation.

L146: p-value is missing in the Table 3.

Table 4 and table 5, Please add Unit.

L159: change to “iron content in sow”.

The discussion is well written, could you please explain the different results between LF and other two iron supplementations? Why the expression of SOD is significantly increased in the heart and reduced in the liver of LF group?

L334-445: The sentence should be described clearly.

L365: Fe-Gly is an inorganic iron? It is a kind of organic iron, please check it carefully.

Author Response

Few studies have compared the effects of different iron supplements on pregnant sows and resulting piglets. This manuscript applied three different kinds of iron sources (heme iron, organic iron, and LF) in sow production and newborn metabolism. It highlights iron supplementation could have a positive effect on reproductive performance in sows. Interestingly, supplementation of LF in pregnant sow better additive effects on the antioxidant capacity than the others. In general, the study revealed important choices of iron source and potentially beneficial role of iron for the metabolism of sow and offspring.

  1. In this experiment, why do you choose heart and liver as the expression tissues of antioxidant genes?

Re: Liver is the main organ that produces antioxidant function, and many experiments related to antioxidant and oxidative damage have chosen liver as the detection site of gene expression. Cardiovascular system is also closely related to oxidative damage. Oxidative stress and inflammatory reaction can cause heart diseases, and the change of antioxidant gene expression level in this part may affect the probability of acquiring heart diseases. The detected genes have high expression levels in both tissues.

  1. Are the contents of iron in the diets of the three treatment groups the same? How to determine whether the difference is caused by the kind of iron additive instead of the iron content? Please supplement the composition of lactoferrin used.

Re: The composition of lactoferrin was added in line 109.

The feed formula of this experiment refers to NRC standard and China pig raising standard, and the basic diet contains 80mg/kg of iron. In our previous study, ferrous sulfate, an inorganic iron additive, was used, but it had little effect on newborn piglets. This experiment mainly wanted to compare the effects of two kinds of organic iron and heme iron additives. The control group was established to verify the effects of these three iron additives, and no additional iron was added, so it can also be called iron deficiency group. What is mentioned in the article is the product content (for example, 200mg/kg), which is because sows have different utilization rates of different additives, and it is not appropriate to only consider the consistency of iron content. The addition amount of the three additives is the suitable addition amount of pregnant sows determined by our previous experiments. Heme iron is provided by the Single Stomach Animal Nutrition Laboratory of Yunnan Key Laboratory of Animal Nutrition and Feed, and extracted from pig blood. The purity is 98%, the heme content is 20%, the iron content is 2%, and the remaining components are HGB, peptides, amino acids and so on.

  1. Please supplement the specific determination methods of iron content in various tissues and serum.

Re: The determination method of iron content has been supplemented in line 132.

  1. Discussion 4.2: Please analyze the relationship between piglets and sows. The discussion part failed to connect the relationship between the iron content of piglets and sows.  Refer to: Dan Wan, et al. Maternal dietary supplementation with ferrous N-carbamylglycinate chelate affects sow reproductive performance and iron status of neonatal piglets. Animal, 2018

Re: L341-L361, this discussion has been revised.

L22: The three additives improved iron nutrition in piglets, with LF and Heme-Fe showing stronger effects. What do you mean? stronger effects on what?

Re: Line 23, changed to “The three additives improved iron nutrition in newborn piglets, with LF and Heme-Fe had better improvement effects.”

Line 42: “an analogous” instead of “a analogous”.

Re: Line 51, changed to “an analogous”

Line 42: “a necessary” instead of “an necessary”.

Re: Line 51, changed to “a necessary”

Line 54: “a high” instead of “high”.

Re: Line 58, changed to “with a high”

Line 63: “disorders” instead of “disorder”.

Re: Line 68, changed to “disorders”

  1. Line 142: “in the LF, Fe-Gly, and Heme-Fe” instead of “in the LF, Fe-Gly, Heme-Fe”.

Re: Line 170, changed to “and 6.09% in the LF, Fe-Gly, and Heme-Fe groups, respectively.”

L117: how did you choose the sacrificed newborn piglets in each group? According to body weight? describe it in detail.

Re: Line 127, changed to “In total, 12 healthy newborn piglets (similar in weight, about 1.5kg) were sacrificed in each group,”

L118: In sample collection part, the collection of breast milk, duodenum, jejunum and ileum is missing.

Re: Line 129, changed to “after which heart, liver, spleen, and lung, duodenum, jejunum and ileum samples were collected within 15 min and frozen in liquid nitrogen at −80 ℃ for gene expression determination.”

L131: what is GRX1 and TRX1? Please add the gene annotation.

Re: The information of primers has been added in line 149.

L146: p-value is missing in the Table 3.

Re: p-value of each item has been added in Table 3.

Table 4 and table 5, Please add Unit.

Re: The unit of each project has been added in Table 3, Table 4 and Table 5.

L159: change to “iron content in sow”.

Re: Line 187, changed to “Effects of different iron additives on iron content in sow”

The discussion is well written, could you please explain the different results between LF and other two iron supplementations? Why the expression of SOD is significantly increased in the heart and reduced in the liver of LF group?

Re: Line 413-418, added “In fact, the expression level of SOD gene in the liver of the treatment group decreased to varying degrees, but it increased to varying degrees in the heart, which may be caused by the change of SOD level in the blood, the maintenance of balance in the body, or the use of iron additives caused a certain load on the liver of newborn piglets, but no liver diseases were observed in piglets in the same litter (not slaughtered, raised on the farm).”

L334-445: The sentence should be described clearly.

Re: The discussion and conclusion section has been revised and rewritten.

L365: Fe-Gly is an inorganic iron? It is a kind of organic iron, please check it carefully.

Re: Line 440, changed to “Fe-Gly is an organic iron”

Reviewer 3 Report

The topic of the present study is interesting and timely because in pigs, iron deficiency anemia is the most prevalent deficiency disorder during the early postnatal period, and must be corrected by iron supplementation. Several studies have attempted to increase the level of iron hepatic iron stores in fetuses by treating pregnant sows with iron supplements. However, supplementation of sows at various stages of pregnancy, using various iron supplements administered orally or parenterally has no significant impact on the improvement of the iron status of newborn piglets and thus does not prevent suckling animals from becoming anemic. However the following issues that arose during the evaluation of the manuscript should be addressed.  

Experimental desigen

In the present study Authors measured iron content in biological samples from sows and piglets, however in the Materials and Methods there is no description of the method  of iron analysis. It is also unclear what was measured total iron, heme iron or no-heme iron ???

The second and more important question is about the final concentration of iron in the diet of the supplemented groups of sows.

Page 3. (L. 100-103).  The Authors claim that “The control group was fed 100 the basic diet, whereas the treatment groups were fed the basal diet supplemented with 200 mg/kg lactoferrin (LF group), 0.8% heme iron (Heme-Fe group), or 500 mg/kg iron-glycine complex (Fe-Gly group)”. It this is a very imprecise statement of the iron content in the diet.

Because Authors analyse the effects of various iron supplements therefore it is necessary to add precise information about the final Fe concentration in mg/kg of diet.

In the present study Authors consider using the lactoferrin as an iron supplement it is very controversial for me, because lactoferrin is a large globular glycoprotein with a molecular mass about 80 kDa protein and one molecule of lactoferrin can bind 2 atoms of Fe. In Abstract of the present paper the Authors claim that “40 pregnant sows were divided into four groups, 18 while sows in the treatment groups were fed diets supplemented with 200 mg/kg of Fe from lactoferrin (LF group)…….” How it is possible ??????

What is more in the Materials and methods I found information that “whereas the treatment groups were fed the basal diet supplemented with 200 mg/kg lactoferrin (LF group),….”  

Please add the information about how many feed in kg was consumed by sow daily.

Please add the information about the method of piglets euthanasia.

Results

In the Results section in the Table 4 and 5 Authors showed the results of analysis of iron concentration in tissue, serum and milk but without any units !!!! Please improve it.

Page 5. L. 150 In the text I found that “According to Table 4, adding iron additives in diet enhanced the iron deposition in 150 sow placenta (p < 0.05).” However, how it is shown in the Table 4, p is < 0,001.

Similar problem with p value is for data presented in Tables 5, 6 and 7.

Page 8. Analysis of gene expression level. Which gene/genes was used as a house keeper/s gene for normalization the expression level of the studied GSH-Px, GRX1, SOD and TRX1 genes.

Author Response

The topic of the present study is interesting and timely because in pigs, iron deficiency anemia is the most prevalent deficiency disorder during the early postnatal period, and must be corrected by iron supplementation. Several studies have attempted to increase the level of iron hepatic iron stores in fetuses by treating pregnant sows with iron supplements. However, supplementation of sows at various stages of pregnancy, using various iron supplements administered orally or parenterally has no significant impact on the improvement of the iron status of newborn piglets and thus does not prevent suckling animals from becoming anemic. However the following issues that arose during the evaluation of the manuscript should be addressed.

Experimental desigen

In the present study Authors measured iron content in biological samples from sows and piglets, however in the Materials and Methods there is no description of the method  of iron analysis. It is also unclear what was measured total iron, heme iron or no-heme iron ???

Re: The determination method of iron content has been supplemented in line 132. The total iron content was determined in the experiment.

The second and more important question is about the final concentration of iron in the diet of the supplemented groups of sows.

Page 3. (L. 100-103).  The Authors claim that “The control group was fed 100 the basic diet, whereas the treatment groups were fed the basal diet supplemented with 200 mg/kg lactoferrin (LF group), 0.8% heme iron (Heme-Fe group), or 500 mg/kg iron-glycine complex (Fe-Gly group)”. It this is a very imprecise statement of the iron content in the diet.

Because Authors analyse the effects of various iron supplements therefore it is necessary to add precise information about the final Fe concentration in mg/kg of diet.

Re: The feed formula of this experiment refers to NRC standard and China pig raising standard, and the basic diet contains 80mg/kg of iron. In our previous study, ferrous sulfate, an inorganic iron additive, was used, but it had little effect on newborn piglets. This experiment mainly wanted to compare the effects of two kinds of organic iron and heme iron additives. The control group was established to verify the effects of these three iron additives, and no additional iron was added, so it can also be called iron deficiency group. What is mentioned in the article is the product content (for example, 200mg/kg), which is because sows have different utilization rates of different additives, and it is not appropriate to only consider the consistency of iron content. The addition amount of the three additives is the suitable addition amount of pregnant sows determined by our previous experiments. Heme iron is provided by the Single Stomach Animal Nutrition Laboratory of Yunnan Key Laboratory of Animal Nutrition and Feed, and extracted from pig blood. The purity is 98%, the heme content is 20%, the iron content is 2%, and the remaining components are HGB, peptides, amino acids and so on.

In the present study Authors consider using the lactoferrin as an iron supplement it is very controversial for me, because lactoferrin is a large globular glycoprotein with a molecular mass about 80 kDa protein and one molecule of lactoferrin can bind 2 atoms of Fe. In Abstract of the present paper the Authors claim that “40 pregnant sows were divided into four groups, 18 while sows in the treatment groups were fed diets supplemented with 200 mg/kg of Fe from lactoferrin (LF group)…….” How it is possible ??????What is more in the Materials and methods I found information that “whereas the treatment groups were fed the basal diet supplemented with 200 mg/kg lactoferrin (LF group),….”  

Re: When designing the experiment, we thought that it was not appropriate to simply unify the content of iron molecules. Different additives have different utilization rates and appropriate addition levels, and they will not be completely consistent. We used the addition amount suitable for pregnant sows of this pig breed obtained through previous experiments. The abstract has been changed to "40 pregnant sows were divided into four groups, the control group was fed basic diet, while sows in the treatment groups were fed diets supplemented with 200 mg/kg lactoferrin (LF group), 0.8% heme-iron (Heme-Fe group), or 500 mg/kg iron-glycine complex (Fe-Gly group)." In the previous article, "18" is the number of the line.

Please add the information about how many feed in kg was consumed by sow daily.

Re: Line 101, added “Pregnant sows were fed 3.5kg of feed every day.”

Please add the information about the method of piglets euthanasia.

Re: Line 128, added “syncope caused by blunt instrument hitting the head, followed by immediate blood-letting,”

Results

In the Results section in the Table 4 and 5 Authors showed the results of analysis of iron concentration in tissue, serum and milk but without any units !!!! Please improve it.

Re: The unit of each project has been added in Table 3, Table 4 and Table 5.

Page 5. L. 150 In the text I found that “According to Table 4, adding iron additives in diet enhanced the iron deposition in 150 sow placenta (p < 0.05).” However, how it is shown in the Table 4, p is < 0,001.

Similar problem with p value is for data presented in Tables 5, 6 and 7.

Re: "P < 0,001" is the overall coefficient of variation of one-way ANOVA, which is 0.05 in multiple comparisons, and is marked with a corner mark in the table, and a footnote has been added. The results have been revised according to the coefficient of variation in multiple comparisons.

Page 8. Analysis of gene expression level. Which gene/genes was used as a house keeper/s gene for normalization the expression level of the studied GSH-Px, GRX1, SOD and TRX1 genes.

Re: Line 153, added “According to the expression of β -actin gene, the expression level of mRNA was standardized, and the relative expression levels were calculated by using the 2−∆∆Ct method [56].”

Round 2

Reviewer 1 Report

Dear authors,

The manuscript was improved after the proposed modification. There are some aspects that were not observed and should be re-worked. 

- According to the NRC (2012), gestating and lactating sows should be supplemented with 80 mg/kg of iron (independently of iron levels from feed ingredients). The present control diet contains 80 mg/kg of Fe from ingredients only. Therefore, the basal diet without iron supplementation is not in accordance with the NRC (2012). Please, correct in the text or provide an explanation supporting your decision to keep it.

- In the methodology, please, clearly state that treatments have similar concentrations of elemental iron. For example: "... whereas the treatment groups were fed the basal diet supplemented with 200 mg/kg lactoferrin (LF group), 0.8% heme iron (Heme-Fe group), or 500 mg/kg iron-glycine complex (Fe-Gly group), which provided similar Fe concentrations between diets.

- The description of breast milk collection should be reviewed as it appears to a more "operational" description than a scientific methodology.

- Table 3: Please, remove "child/only" and replace by "n".

- Item 3.5: Please, remove "the highest in 522 the LF group" in the first line and "the lowest was the LF group" in the last line of this item.

Figure 1: Please, define LF and remove "Same below"

Author Response

Reply to Reviewer 1

- According to the NRC (2012), gestating and lactating sows should be supplemented with 80 mg/kg of iron (independently of iron levels from feed ingredients). The present control diet contains 80 mg/kg of Fe from ingredients only. Therefore, the basal diet without iron supplementation is not in accordance with the NRC (2012). Please, correct in the text or provide an explanation supporting your decision to keep it.

Re: The “control group” in the manuscript has been changed to “iron deficiency group” and redefined.

- In the methodology, please, clearly state that treatments have similar concentrations of elemental iron. For example: "... whereas the treatment groups were fed the basal diet supplemented with 200 mg/kg lactoferrin (LF group), 0.8% heme iron (Heme-Fe group), or 500 mg/kg iron-glycine complex (Fe-Gly group), which provided similar Fe concentrations between diets.

Re: Line 109, added “and the iron available to sows in the treatment groups was similar.”

Re: Line 302-304, added “The bioavailability of iron from different sources is not the same [57]. According to the previous experiments, we selected iron additives suitable for this study and their dosage, and they can provide similar iron concentrations for sows [60].”

- The description of breast milk collection should be reviewed as it appears to a more "operational" description than a scientific methodology.

Re: Line 126-132, changed to “Breast milk samples were collected on the first day, the second day and the third day after delivery. disinfect nipples before collection, select 3-4 pairs of nipples before navel, and put a single nipple into a sterilized 50ml centrifuge tube, gently squeezed the nipple root to collect breast milk in the centrifuge tube, collected 10-15ml from each nipple, mixed the breast milk collected from each nipple evenly. Mixed the breast milk collected by each pig for three days evenly, stored at −20 ℃.”

- Table 3: Please, remove "child/only" and replace by "n".

Re: Table 3, changed to “n”.

- Item 3.5: Please, remove "the highest in 522 the LF group" in the first line and "the lowest was the LF group" in the last line of this item.

Re: Has been deleted.

Figure 1: Please, define LF and remove "Same below"

Re: Figures 1 and 2, changed to “LF: lactoferrin group” and deleted “Same below”.

Reviewer 3 Report

In the present form manuscript was significantly improved by Authors, especially the methodological section. However my question about final iron concentration in mg/kg in the diet of heme-Fe group and Gly-Fe group is still open. Because according to the good scientific practice it is possible to compare the effect of iron supplementation between two groups if sows received the same amount of this element in the diet.  It is interesting, and should be also shown because Authors found  increasing of iron concentration in the organs of the piglets, however many literature data indicated that oral supplementation of pregnant sows with heme- or ionic-iron is an ineffective form of therapy.  Thus, results of the present study should be very clearly documented.

Author Response

Reply to Reviewer 3

In the present form manuscript was significantly improved by Authors, especially the methodological section. However my question about final iron concentration in mg/kg in the diet of heme-Fe group and Gly-Fe group is still open. Because according to the good scientific practice it is possible to compare the effect of iron supplementation between two groups if sows received the same amount of this element in the diet.  It is interesting, and should be also shown because Authors found increasing of iron concentration in the organs of the piglets, however many literature data indicated that oral supplementation of pregnant sows with heme- or ionic-iron is an ineffective form of therapy.  Thus, results of the present study should be very clearly documented.

Re: Line 109, added “and the iron available to sows in the treatment groups was similar.”.

Line 302-304, added “The bioavailability of iron from different sources is not the same [57]. According to the previous experiments, we selected iron additives suitable for this study and their dosage, and they can provide similar iron concentrations for sows [60].”.

Line 348-353, added “Heme iron, as an iron supplement to treat human iron deficiency, was rarely used to treat pregnant sows at present [58]. Medical research shows that for pregnant women, heme iron was a dietary iron source with higher bioavailability than FeSO4. [61] Heme iron used in this study was extracted from pig blood by our research group, which showed good bioavailability in the previous experiments [59].”.

There are few reports on the effect of adding heme iron to sow diet on newborn piglets. Moreover, heme iron has also been proved to have high bioavailability in pregnant women. Through this study, we also found that heme iron is different from the other two iron supplements, and the reasons for this phenomenon will be further studied in the future.
